# Effects of different aerobic exercises on the global cognitive function of the elderly with mild cognitive impairment: a meta-analysis

Conglin Han [1,2] Weishuang Sun,[1,2] Dan Zhang,[1,2] Xiaoshuang Xi,[3] Rong Zhang,[4] Weijun Gong [5]

[1]Beijing Rehabilitation Hospital, Capital Medical University, Beijing, China
[2]Rehabilitation Medicine Academy, Weifang Medical University, Weifang, Shandong, China
[3]Beijing Rehabilitation Medicine Academy, Capital Medical University, Beijing, China
[4]Second Clinical Medical Academy, Yunnan University of Chinese Medicine, Kunming, Yunnan, China
[5]Department of Neurological Rehabilitation, Beijing Rehabilitation Hospital, Beijing, China

**Correspondence to**
Dr Weijun Gong;
gwj197104@ccmu.edu.cn

## ABSTRACT

**Objectives** To summarise the effects of various types of aerobic exercise on the global cognitive function of the elderly with mild cognitive impairment (MCI).

**Design** A meta-analysis of randomised controlled trials (RCTs).

**Data sources** PubMed, EMBASE and the Cochrane Library were searched for clinical RCTs from the earliest available records to March 2022.

**Eligibility criteria for selecting studies** We included RCTs of subjects older than 60 years with MCI. The outcome indicators of cognitive function of interest were the Mini-Mental State Examination (MMSE) and the Montreal Cognitive Assessment (MoCA).

**Data extraction and synthesis** Two researchers independently screened the literature, extracted data and evaluated the quality of the included studies, with disagreements resolved by a third researcher. The *Cochrane Handbook for Systematic Reviews of Interventions* was used to assess the risk of bias. Meta-analysis was performed by Review Manager V.5.3 software. Random-effect models were used for meta-analysis.

**Results** A total of 1680 patients who participated in 20 RCTs were included in this study. Based on outcomes of MMSE analysis, the aerobic exercise, which was beneficial for global cognitive function in MCI patients, included multicomponent aerobic exercise (MD=1.79, 95% CI (1.41 to 2.17), p<0.01) and mind–body exercise (MD=1.28, 95% CI (0.83 to 1.74), p<0.01). The results of the meta-analysis of conventional aerobic exercise (MD=0.51, 95% CI (0.09 to 0.93), p=0.02) turned out to be statistically insignificant after sensitivity analysis (MD=0.14, 95% CI (−0.47 to 0.75), p=0.65). With the evaluation of MoCA, multicomponent aerobic exercise (MD=5.74, 95% CI (5.02 to 6.46), p<0.01), mind–body exercise (MD=1.29, 95% CI (0.67 to 1.90), p<0.01) and conventional aerobic exercise (MD=2.06, 95% CI (1.46 to 2.65), p<0.01) were showed significant beneficial effects for the patient. However, there was a high degree of heterogeneity between the results of multicomponent aerobic exercise (MMSE) and conventional aerobic exercise (MoCA), which was analysed and explored.

**Conclusions** In general, multicomponent aerobic exercise and mind–body exercise were beneficial in improving global cognitive function in the elderly with

## STRENGTHS AND LIMITATIONS OF THIS STUDY

⇒ A rigorous methodology was used for this study, including clear eligibility criteria, an extensive database search, study selection by two reviewers working independently and risk-of-bias assessment.
⇒ This protocol is based on the Preferred Reporting Items for Systematic Reviews and Meta-Analyses project and the *Cochrane Handbook for Systematic Reviews of Interventions*.
⇒ This study is the first to classify aerobic exercise and conduct a meta-analysis to explore the effects of different aerobic exercises on cognitive function in older adults with mild cognitive impairment (MCI).
⇒ This meta-analysis will be used to inform future research to guide and improve aerobic exercise training practices in older adults with MCI.
⇒ This study only analysed changes in global cognitive functioning and did not include changes in specific cognitive domains.

MCI. Nevertheless, the improvement effect of mind–body exercise is more reliable compared with multicomponent aerobic exercise and conventional aerobic exercise.

**PROSPERO registration number** CRD42022327386.

## INTRODUCTION

Along with the development and improvement in modern science, medical technology and treatment, the average life expectancy of people around the world has been continuously extended. Age-related health problems are becoming more and more prominent. Dementia is one of the common diseases of ageing and a major cause of disability in the elderly.[1] The WHO estimates that about 50 million people worldwide suffer from dementia, with a new case every 3 s, and the number of people with dementia will be three times by 2050.[2] Severe dementia can place a huge emotional and financial burden on families and society, so it is important to

take measures for early detection, early intervention and early treatment of dementia. Mild cognitive impairment (MCI) is an intermediate state between normal cognition and dementia. MCI usually refers to cognitive decline beyond that caused by normal ageing, but without significant impairment in activity of daily living (ADL).[3] Currently, medications have a limited role in preventing and treating MCI,[4 5] so there is a large body of research focusing on non-pharmacological interventions.

It has been suggested that some non-pharmacological interventions, such as food, exercise, and cognitive stimulation, may be beneficial for the improvement of cognitive function in MCI patients.[3 4 6] It has also shown that physical activity has a greater influence on the cognitive performance of MCI patients than drugs. For the elderly population with a decline in cognitive function, several studies have also confirmed that aerobic exercise appears to be effective in improving cognitive function.[7–9] However, there are various types of aerobic exercise, and the aerobic exercise that can be performed may vary in different populations (for example, it is difficult for people with lower limb mobility restrictions to run). As a result, an analysis is needed to compare the effects of different types of aerobic exercise on the improvement of global cognitive function in the elderly with MCI.

## METHODS

This study has been registered in the International Prospective Systematic Evaluation Registry database (PROSPERO, http://www.cdr.york.ac.uk/PROSPERO/) under the registration number: CRD42022327386. This study was conducted following the Preferred Reporting Items for Systematic Reviews and Meta-Analyses statement.

### Inclusion criteria
#### Types of studies
Human subject studies designed as randomised controlled trials (RCTs) published in English were included without year restriction.

#### Types of participants
Subjects defined as MCI by the original authors and aged>60 years were included. We excluded healthy ageing elderly adults, patients with any form of dementia or those diagnosed with cognitive impairment due to definite aetiologies, such as trauma or vascular or psychiatric diseases. Any clinical subtype of MCI was considered eligible.

#### Types of interventions
We included supervised structured exercises of any frequency, intensity, duration, or time directed at improving physical fitness. The interventions were various forms of exercise containing aerobic exercise, including conventional aerobic exercise (simple and common aerobic exercises such as running, brisk walking and cycling), mind–body exercise (mind–body exercise focuses on mind, body, psychology and behaviour,

including breathing and physical exercise, meditation and so on.[10] It is a type of aerobic exercise of low to moderate intensity, with common forms such as tai chi, yoga and dance.[11 12]) and multicomponent aerobic exercise (aerobic exercise combined with other forms of exercise, such as resistance training, balance training). Aerobic exercise was defined as the involvement of major muscle groups throughout the body in addition to the primary involvement of oxygen for energy supply.

#### Types of control groups
Controls could be exercises of stretching, activities of health education, routine care, daily lifestyle, and social recreation.

#### Types of outcomes
The results are presented using measurable cognitive screening instruments: the Mini-Mental State Examination (MMSE) and the Montreal Cognitive Assessment (MoCA). and are available as mean (M) and SD.

### Exclusion criteria
Intervention methods could only be a variety of exercise training modalities; those with specific instructions to have other training components, such as cognitive training, were excluded. Non-intervention studies, articles with only research protocols, review articles, unpublished studies, abstracts or papers, and articles for which full text or data were not available were excluded. Non-English articles were excluded.

### Search strategy
We systematically searched electronic databases in PubMed, EMBASE and the Cochrane Library from the earliest available records to March 2022. We searched using subject terms and free words. The keywords used included search terms related to sports exercise (eg, exercise or physical activity or aerobic exercise) and cognitive impairment (eg, or cognitive dysfunction or amnestic MCI) and search terms related to older adults (eg, older adults or older patients). The search strategies for EMBASE, PubMed and the Cochrane Library are described in online supplemental document 1. Endnote X9.1 was used to store and sort the retrieved RCTs and to remove duplicates. Initial screening was performed by the first author based on the title and abstract. This was followed by an independent screening of the full text by two reviewers according to the eligibility criteria specified in the study protocol. Disagreements in included studies were resolved by negotiation.

### Data extraction and analysis
A standardized data extraction form was applied to extract data based on the following parameters: (1) basic information about the included studies (author, year, country), (2) basic characteristics of the study population (sample size, age and gender of participants), (3) intervention characteristics (type of intervention, the intervention specifics, frequency and intensity of the intervention)

and (4) outcome measures: we chose the two most commonly used scales for assessing global cognitive functioning, the MMSE and the MoCA. We performed a meta-analysis using the endpoint measures. First, we conducted a meta-analysis of all aerobic exercise studies to obtain the results of the meta-analysis of MMSE and MoCA. After that, according to the type of aerobic exercise, they were divided into three groups (multicomponent aerobic exercise, mind–body exercise and conventional aerobic exercise) and meta-analysis was conducted separately. For each type of aerobic exercise, meta-analysis results of MMSE and MoCA were also obtained.

For each outcome, the effect sizes are reported as the mean difference (MD) with 95% CI for each study. We pooled these values and conducted the meta-analysis using the Review Manager V.5.3. Heterogeneity between trial design options was unclear; therefore, a random effects model was chosen. Our random-effect model was based on the postulation that effect estimates were not the same but followed a normal distribution. While the area of the black square in forest plots denotes the weighted contribution of each study, $p < 0.05$ (two tailed) was considered statistically significant. The $I^2$ statistic was computed as a measure of heterogeneity.[13] Cochrane's handbook recommends heterogeneity as not important (0%–40%), moderate heterogeneity (30%–60%), substantial heterogeneity (50%–90%) and considerable heterogeneity (75%–100%). Sensitivity analysis evaluated the stability of the results by leave-one-out method. Subgroup analyses were used to address significant clinical heterogeneity, or only descriptive analyses were performed.

### Risk of bias and quality assessment

The Cochrane risk of-bias assessment tool was used to independently assess the risk of bias in included trials. The assessment covered the following parameters: random sequence generation, allocation concealment, blinding, incomplete data, selective reporting and others (*Cochrane Handbook for Systematic Reviews of Interventions* 8.5). Each parameter was divided into three categories: low risk, unclear and high risk. Each study was categorised using the following criteria: low risk of bias (all parameters were rated as low risk), moderate risk of bias (with parameters rated as unclear and no parameters rated as high risk) and high risk of bias (as long as one or more parameters were rated as high risk) (*Cochrane Handbook for Systematic Reviews of Interventions* table 8.7.a). Two reviewers (CH and WS) assessed each trial independently. Any disagreements were discussed with a third reviewer (WG) until a final decision was achieved.

### Assessment of evidence quality

The overall quality of the evidence was assessed using the Recommendations for Assessment, Development and Evaluation (GRADE) criteria.[14] In the system, the quality of evidence is classified into four levels: high quality, moderate quality, low quality and very low quality.[14]

### Patient and public involvement

There was no direct patient or public involvement in this review.

## RESULTS

### Search results

By searching electronic databases and literature, we obtained a total of 24 255 relevant references. After duplicates were removed, and titles and abstracts were checked against predefined inclusion criteria, 102 complete references were retained, and we reviewed each paper in detail. A total of 84 trials did not meet the predefined inclusion criteria: non-RCTs (n=4), no control group meeting the criteria (n=5), full text not available (n=6), assessment results not meeting the criteria (n=23), the study without results (n=9), data not available (n=13), intervention modality not meeting the predefined criteria (n=7) and study population not meeting the inclusion criteria (n=17). Therefore, a total of 18 articles were included in this study. The search process is shown in figure 1.

### Research characteristics

A total of 18 RCTs were included in this study, published from 2011 to 2022. The characteristics of all included studies are summarised in online supplemental document 2 . The studies were conducted in different countries: China (n=8), the USA (n=2), Brazil (n=2), Japan (n=1), Korea (n=1), Turkey (n=1), Iran (n=1), Greece (n=1) and Spain (n=1). All included participants were 1700 and they were assigned to either the experimental or control group, and the sample size of the study ranged from 19 to 389. All participants were diagnosed with MCI. One study[15] was composed of the mind–body exercise intervention group and the conventional aerobic intervention group with their corresponding control group. Another study[16] was designed as two groups with different exercise intensities (40% and 60% reserve heart rate) for the conventional aerobic intervention. For all included studies, five experimental groups underwent multicomponent exercise training with aerobic exercise,[17–21] seven experimental groups underwent mind–body exercise training[15 22–27]; and eight experimental groups underwent conventional aerobic exercise training.[15 16 28–32] The total intervention duration for all studies ranged from 6 weeks to 12 months. The frequency was 21–80 min per session, 1–5 times per week. Three experimental groups had an exercise intensity of roughly low intensity,[15–17] that is, maximum heart rate<60% or Borg scale (level 10)<5 or Borg scale (level 20)<12 or metabolic equivalent during exercise (MET)<3 or reserve heart rate<59%; 15 experimental groups had an exercise intensity of roughly moderate to high intensity,[15 16 18–20 22–29 31 32] that is, maximum heart rate≥60% or Borg scale (10th grade)≥5 or Borg scale (20th grade))≥12 or MET≥3 or reserve heart rate≥59%; the intensity of exercise in two other experimental groups was unclear.[21 30] Ten studies assessed global cognitive function with the MMSE,[16 18–21 26–28 31 32]

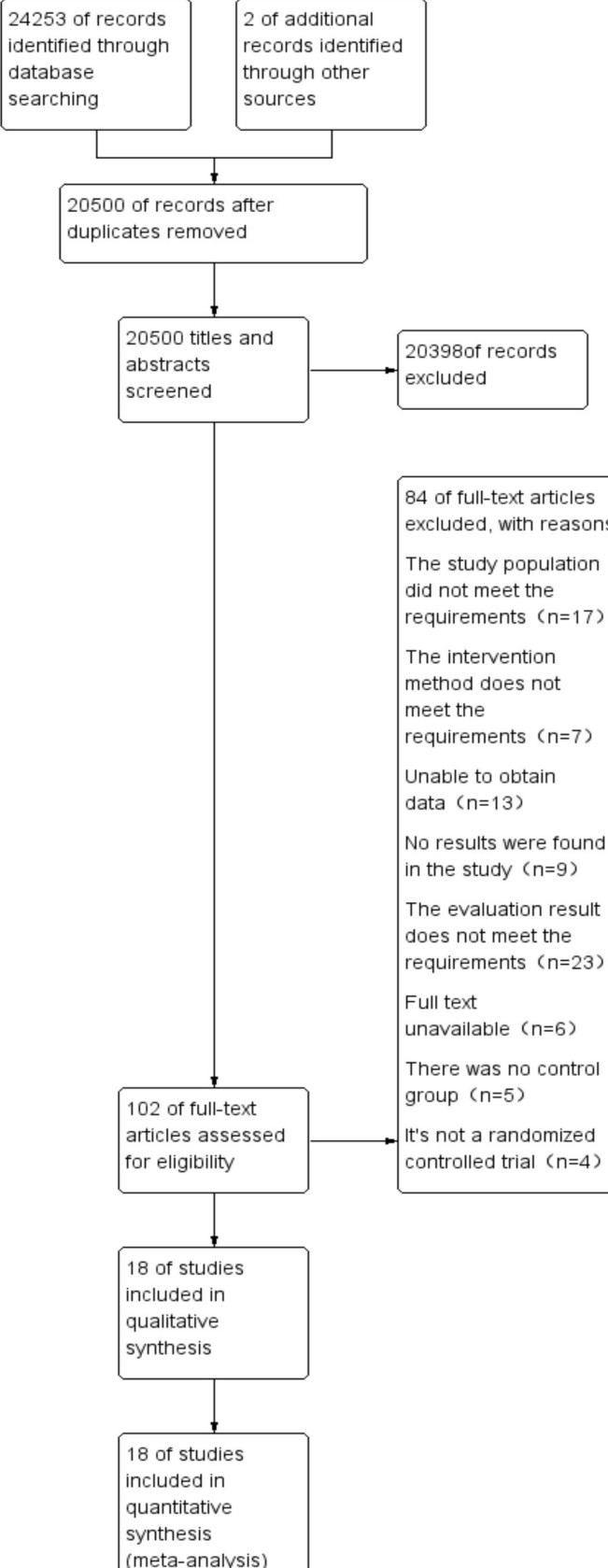

**Figure 1** Study flow diagram.

five studies assessed global cognitive function with the MoCA[15 22 24 29 30] and three additional studies used both the MMSE and the MoCA.[17 23 25]

### Literature quality evaluation

As shown in figure 2, the quality of studies included in the trials varied. Two trials had a low overall risk of bias,[24 29] five trials had a moderate overall risk of bias[15 18 22 27 28] and eleven trials had a high overall risk of bias.[16 17 19–21 23 25 26 30–32] Regarding the method of randomisation, 13 trials[15 17 18 20–25 29–32] had simple randomization and 5 trials[16 19 26–28] had stratified randomisation. In terms of intervention allocation concealment, nine trials[17 19–21 23 25 27 28 30] used open randomisation tables, three trials[22 24 29] used sealed envelopes and the remaining six trials[15 16 18 26 31 32] were unclear.

We conducted a subgroup analysis based on the quality of the studies in order to test the strength of the evidence. The relevant results are shown in online supplemental document 3. Generally, only the moderate risk group of conventional aerobic exercise (MoCA) had statistically insignificant results and there was only one study in this group.

### Results of meta-analysis of all included studies

The results of MMSE included a total of 14 trials, as shown in figure 3. Meta-analysis of the random effects model showed that aerobic exercise was effective in improving global cognitive function compared with the control group (MD=1.23, 95% CI (0.99 to 1.47), p<0.01). Significant heterogeneity was found in the study (I²=88%, p<0.01).

The results of MoCA included a total of eight trials, as shown in figure 3. Randomised effects model meta-analysis showed that aerobic exercise was effective in improving global cognitive function compared with control group (MD=2.94, 95% CI (2.58 to 3.30), p<0.01). Significant heterogeneity was found in the study (I²=94%, p<0.01).

### Results of meta-analysis of different aerobic exercises

All trials were divided into multicomponent aerobic exercise, mind–body exercise and regular aerobic exercise by exercise modality and the results of meta-analysis are shown in figures 4 and 5. As can be seen from the figures, after grouping by exercise modality, the between-group differences in the two outcome indicators of MMSE (p<0.01) and MoCA (p<0.01) were statistically significant, indicating that the grouping was meaningful.

### Results of meta-analysis of multicomponent aerobic exercise

As shown in figure 4, multicomponent aerobic exercise had a significant beneficial effect in the MMSE outcome index (MD=1.79, 95% CI (1.41 to 2.17), p<0.01) but the heterogeneity was very high (I²=95%, p<0.01), so further analysis is needed. We performed a subgroup analysis of the multicomponent intervention group, and the results are shown in online supplemental document 4. We performed a subgroup analysis based on the sample

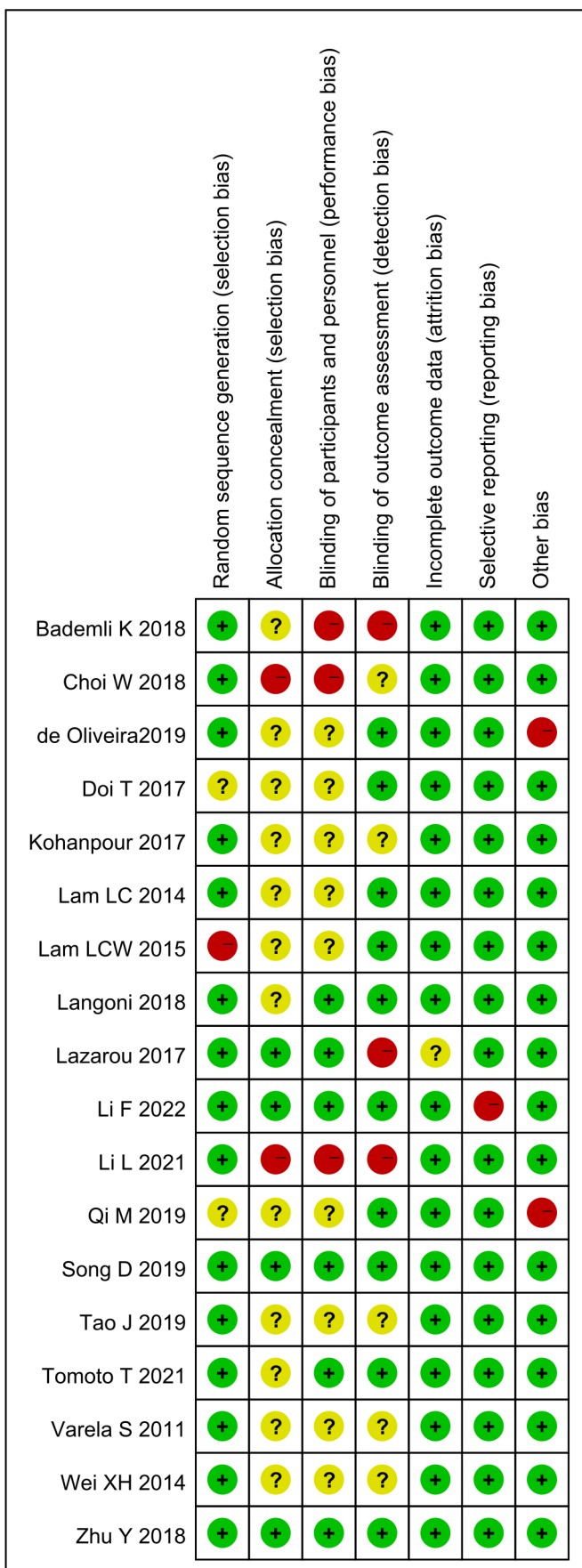

**Figure 2** Risk-of-bias assessment of included trials.

size, exercise intensity, region, deviation risk rating and duration, frequency and total time in the trial. From the results of these analyses, it is clear that no meaningful results were obtained after subgroup analysis. As shown in figure 5, there was only one multicomponent exercise trial with a MoCA outcome and the result was (MD=5.74, 95% CI (5.02 to 6.46), p<0.01).

### Results of meta-analysis of mind–body exercise

As shown in figure 4, in the MMSE outcome, the mind–body exercise intervention had a significant improvement effect (MD=1.28, 95% CI (0.83 to 1.74), p<0.01) and was not highly heterogeneous ($I^2$=50%, p=0.11). As shown in figure 5, there was also a clinically significant effect of mind–body exercise in the results of MoCA (MD=1.29, 95% CI (0.67 to 1.90), p<0.01) and its heterogeneity was also low ($I^2$=6%, p=0.36).

### Results of meta-analysis of conventional aerobic exercise

As shown in figure 4, in the outcome of MMSE, the improvement in global cognitive function was significant in the conventional aerobic exercise intervention group (MD=0.51, 95% CI (0.09 to 0.93), p=0.02) and without heterogeneity ($I^2$=0%, p=0.46). Figure 5 demonstrates the MoCA results for conventional aerobic exercise. The results are also significant (MD=2.06, 95% CI (1.46 to 2.65), p<0.01), but the heterogeneity ($I^2$=84%, p=0.002) is high requiring further analysis and interpretation.

### Rating the body of evidence

Online supplemental document 5 presents the GRADE evidence profile for the RCT outcomes. The quality of evidence varies from moderate to very low. For multicomponent exercise, there is very low evidence (MMSE and MoCA) that it has a significant clinical effect on global cognitive function. For mind–body exercise, moderate (MMSE) and low (MoCA) evidence suggests a clinically significant effect on global cognitive function. For conventional aerobic exercise, there is low and very low evidence of a clinically significant effect on global cognitive function.

### Sensitivity analysis

Using a leave-one-out method, we tested the effect of excluding individual studies on the stability of the effect estimates. We found no significant effect on effect estimates after excluding each study, except for the results for conventional aerobic exercise (MMSE). This suggests that the vast majority of the final results were stable. Regarding the MMSE results for conventional aerobic exercise, after sensitivity analysis, we found that the results were no longer significant after excluding the Wei and Ji's study[32] (MD=0.14, 95% CI (-0.47 to 0.75), p=0. 65). The details of the sensitivity analysis are presented in online supplemental document 6.

A

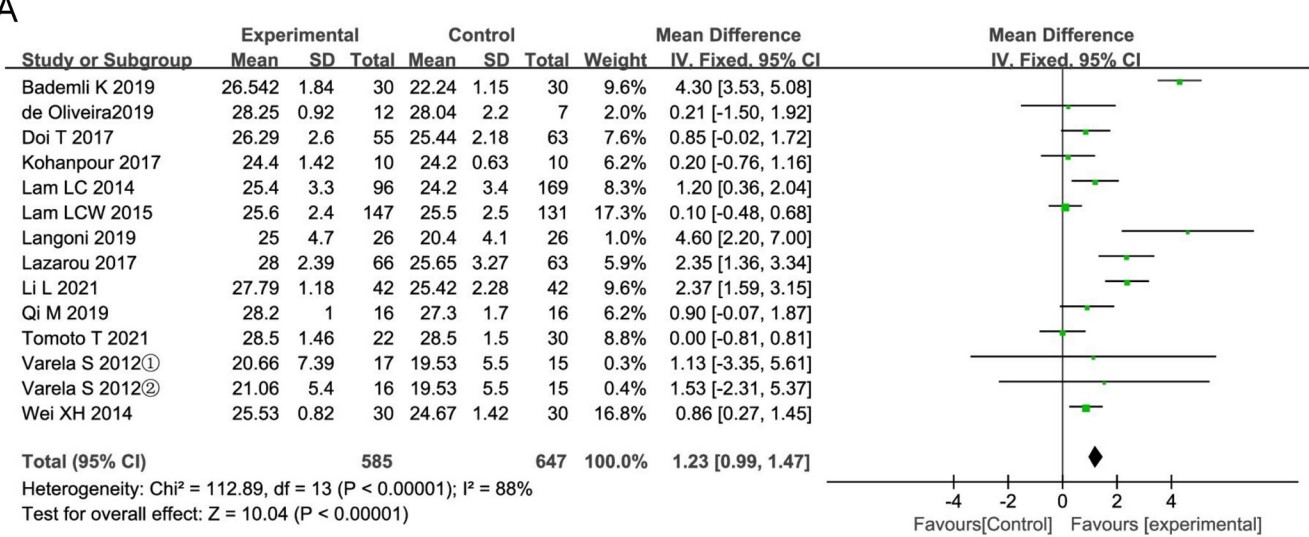

B

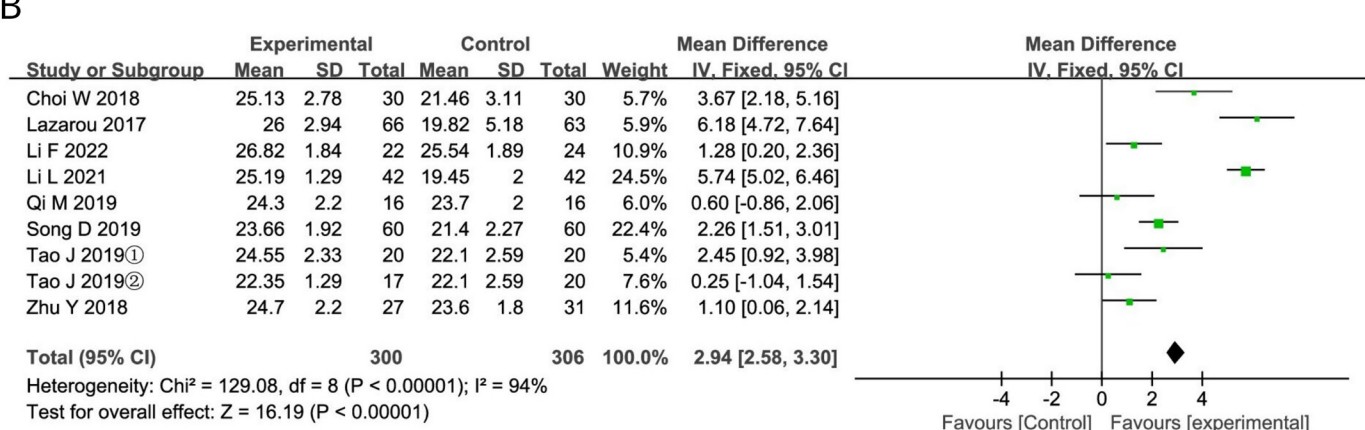

**Figure 3** (A) Effect sizes of forest plots for aerobic exercise versus controls—MMSE outcomes. MMSE, Mini-Mental State Examination. (B) Effect sizes of forest plots for aerobic exercise versus controls—MoCA outcome. MoCA, Montreal Cognitive Assessment.

## DISCUSSION

To the best of our knowledge, our meta-analysis explored for the first time the effect of different aerobic exercise types on global cognitive function in elderly patients with MCI. Our study analysed improvements in cognitive function through endpoint indicators of global cognitive function and found that both multicomponent aerobic exercise and mind–body exercise produced positive effects. In addition, the results of conventional aerobic exercise were inconsistent and may not have a clinically significant effect on global cognitive function. The effects of conventional aerobic exercise need to be further investigated in a large number of high-quality studies.

### Summary of the literature

There were few previous meta-analyses[33 34] that explored the effects of multicomponent exercise on cognitive function in MCI. Although some of their results suggested a beneficial effect of multicomponent aerobic exercise

on cognitive function in the elderly with MCI, the intervention criteria for the studies included in these meta-analyses were not only exercise but also cognitive training and mental stimulation, which were not the same as in our study. The Meta-analysis by Borges -Machado et al[35] discussed multicomponent exercise, which was defined as exercise that combined aerobic, strength, posture and balance training. Although the study found improvements in ADL in the subjects, the study was conducted in patients with dementia and the results did not determine the effects on cognitive function.

Previous studies[36 37] confirmed the beneficial effects of mind–body exercise on global cognitive function in people with MCI. Unlike the present study, they had slight differences in the type of trial studies, age and outcome evaluation indicators in the inclusion criteria. Our inclusion criteria were relatively more stringent. These previous studies combined different cognitive evaluation

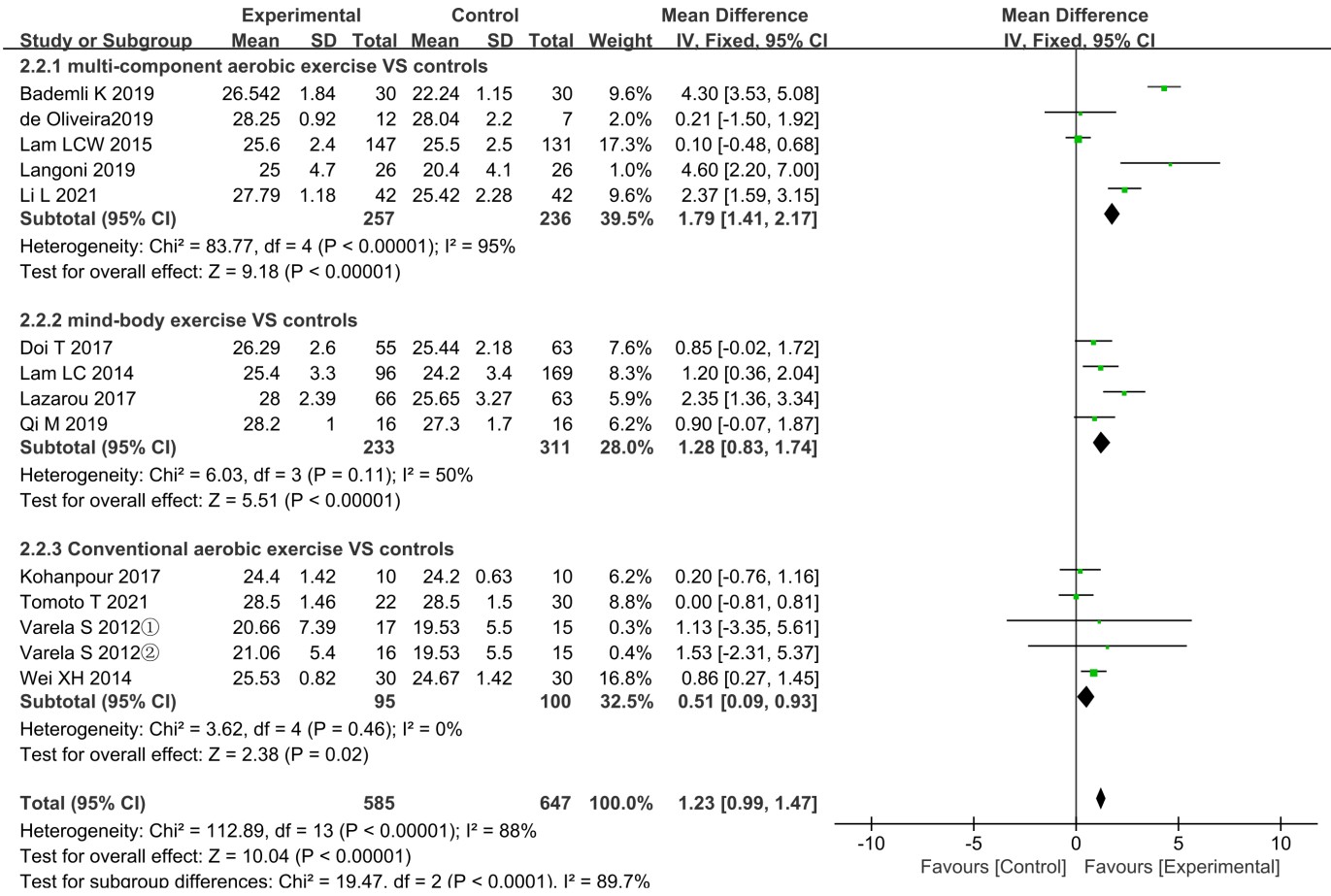

**Figure 4** Effect sizes of forest plots for the different aerobic exercise—MMSE outcomes. MMSE, Mini-Mental State Examination.

metrics expressed as standardised MD. We selected the two most common indicators of global cognitive function evaluation (MMSE and MoCA) to be analysed separately and expressed the results as MD.

We did not find any meta-analysis related to conventional aerobic exercise in patients with MCI. Although there are a few reviews[9 38 39] reporting aerobic exercise in MCI patients, they did not analyse the types of aerobic exercise separately. Therefore, we believed that the analysis of the present study was still of practical relevance.

### Analysis of the results of mind–body exercise

All studies of mind–body exercise reported positive results. The results of both the MMSE and MoCA meta-analyses were significant effects and low heterogeneity, which would seem to indicate the effectiveness of mind–body exercise in improving cognitive function in older adults with MCI. One possible explanation for the beneficial results of the mind–body exercise was that mind–body exercise was a form of physical exercise supplemented with different degrees of cognitive function training. For example, Tai Chi, a Chinese system of slow meditative physical exercise designed for mind relaxation and balance, required attention, memory and other aspects in addition to physical activity. Adding cognitive training to physical activity seems to have better effects on

cognitive function improvement than training alone.[40] Another possible explanation was that depression was an important risk factor for cognitive decline in people with MCI and that mind–body exercise improved mood in older adults better than other physical exercises.[36]

### Heterogeneity analysis of multicomponent aerobic exercise

From a total of five studies analysed by MMSE for multicomponent aerobic exercise, two studies[19 21] were no meaningful change in outcomes. We analysed these studies and speculated that the possible reasons for such high heterogeneity. In the trial studied by de Oliveira *et al*,[19] the authors indicated that participants in both the trial and control groups were receiving medication, but no further medication-related descriptions. We are not sure if the medication affected the cognitive function of the subjects and thus the effect of the intervention. In another trial[21] that reported negative results, each exercise type of its multicomponent aerobic exercise was performed separately, whereas common multicomponent exercises combine different components in a single exercise session.[35] This was therefore the most different aspect of this study from other multicomponent exercise training, and one possible explanation was that different components were more effective when combined in a single exercise session. One study showed[41] that

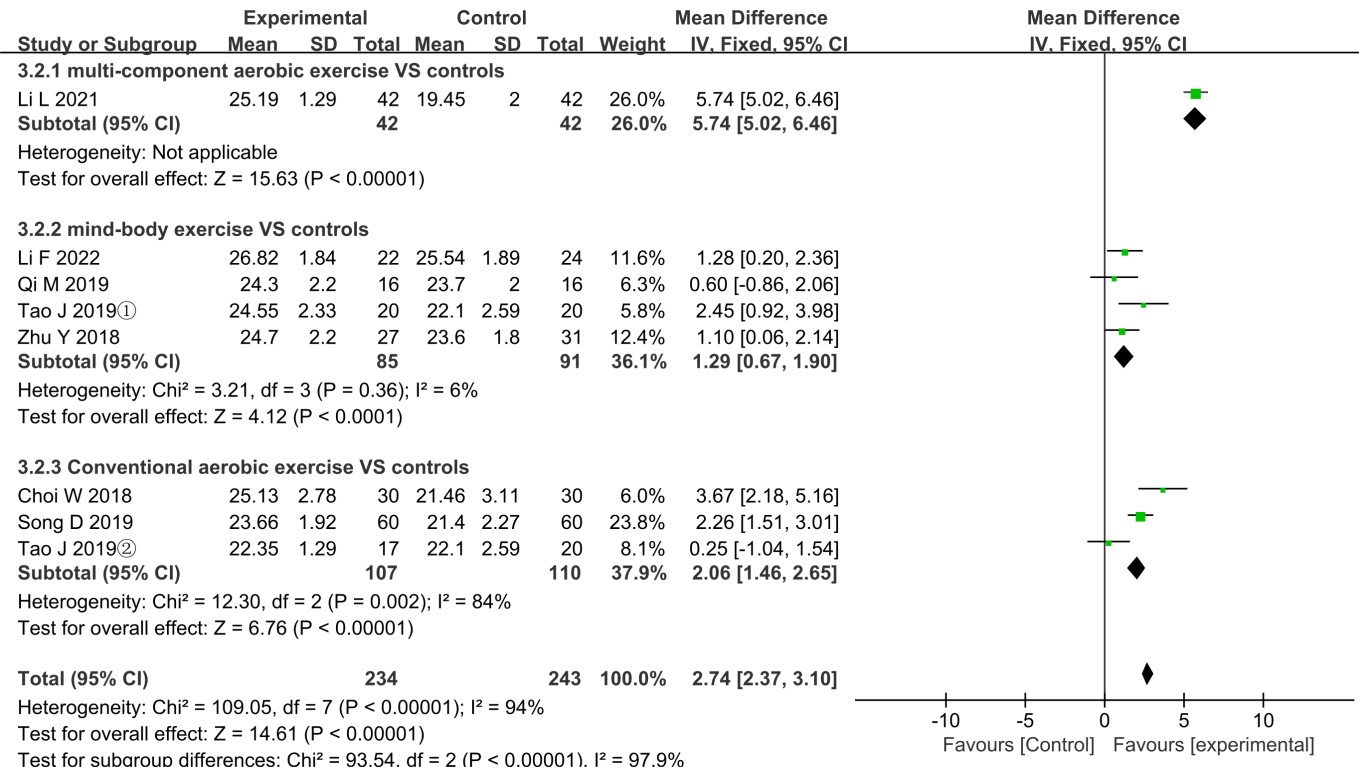

**Figure 5** Effect sizes of forest plots for the different aerobic exercise—MoCA outcomes. MoCA, Montreal Cognitive Assessment.

short-term tai chi training did not significantly improve cognitive function in adults when comparing long-term and short-term tai chi training. In another trial,[17] the duration of aerobic exercise was too short at only 5 min. Thus, we speculated that there may be some difference in the results produced by different time of aerobic exercise.

**Analysis of the results of conventional aerobic exercise**

The preliminary MMSE results for conventional aerobic exercise were significant, which would seem to demonstrate the effectiveness of conventional aerobic exercise. However, after sensitivity analysis was performed to exclude this Wei and Ji's study,[32] this result became meaningless. This suggests that the MMSE results for conventional aerobic exercise are unstable and we cannot conclude the effectiveness of conventional aerobic exercise on this basis. In fact, despite the large number of studies and recommendations[9 42 43] that conclude that aerobic exercise in life is beneficial for cognitive function, there are still many evidence-based studies[44 45] that have yielded non-positive results. A 4-year trial[46] showed that aerobic exercise alone was not sufficient to change cognitive function, and that aerobic exercise up to a certain intensity and in combination with other forms of exercise might have better results. This may be the reason why the results are not positive.

The results of the meta-analysis on MoCA produced a high level of heterogeneity, which we tried to explain. First of all, in the three studies[15 29 30] with this outcome, there was a larger proportion of female subjects, which may have made the studies unrepresentative. A previous

study[47] demonstrated a sex bias in older adults regarding the effects of aerobic exercise on cognitive function, with older women experiencing more significant cognitive benefits from aerobic exercise interventions than men. Second, one of the included studies[15] had a small sample size, which may have made the results of the trial less reliable.

In summary, given the above two results, we are cautious about the effect of conventional aerobic exercise. We believe that the improvement in cognitive function with conventional aerobic exercise may not always be significant and that future studies could improve the quality of research and provide more in-depth studies and analyses of specific interventions.

**Limitations**

There were several limitations of this review. The first is the number of included studies. Although the total number of included studies is not small, the interventions primarily explored in this study were a variety of different aerobic exercises, so the small number of included studies for each type of aerobic exercise makes the interpretation of the results obtained rather difficult. Second, the quality of the studies we included varied, and low-quality studies may make the results inaccurate. The third is that there was considerable heterogeneity between some of the included studies, representing both clinical and methodological heterogeneity, due to differences between participants (eg, older adults may have unknown comorbidities and the possibility of taking medications as a result, which may affect the effectiveness

of the intervention), interventions, and study designs. Fourth, it cannot be excluded that certain control groups would also have different levels of benefit on cognitive function such as social activities.[48] Fifth, only some of the trials that assessed global cognitive outcomes were included, and not those that assessed separately cognitive domains. Finally, only studies of trials with MMSE and MoCA outcomes were selected. Although these two cognitive evaluation metrics are the most commonly used, they also undoubtedly excluded other studies that might be of value.

## Implications for future research

Future studies should be designed with rigorous methods as well as controlled for various biases to improve the quality of the study. Different types of aerobic exercise have different experimental interventions. The duration, frequency and intensity of the exercise intervention may have different degrees of influence on the results of the trial. Follow-up research protocols could focus on this to explore the design of better interventions. The exercise components of multicomponent aerobic training are complex and future studies could investigate better combinations of exercise components to achieve better improvements.

In addition, previous studies have indicated that the MoCA is superior to the MMSE for screening diagnoses in MCI patients, making it more recommended to replace the MMSE with the MoCA for future scales measuring global cognitive function.[49] In addition to the MMSE and the MoCA, the Clinical Dementia Rating is a worthwhile scale to refer to and use. In addition to the global cognitive function, it would be interesting to assess specific domains of cognitive function (eg, memory, attention), which may allow the inclusion of additional studies and better delineate the sources of heterogeneity involved.

## CONCLUSION

The results of this meta-analysis suggest that two types of aerobic exercise (including multicomponent aerobic exercise and mind–body exercise) have a positive effect on improving global cognitive function in older patients with MCI. However, the conclusion should be treated cautiously due to the heterogeneity and limited research. Additional well-designed RCTs are needed to provide more evidence for this conclusion.

**Contributors** All authors made a substantial contribution to this work. CH, WS, DZ, XX, RZ and WG all contributed to the conception and design of the review. CH drafted the paper with all the authors critically reviewing it and suggesting amendments prior to submission. CH and WG are guarantors. The corresponding author attests that all listed authors meet authorship criteria and that no others meeting the criteria have been omitted.

**Funding** This work was supported by Capital's Funds for Health Improvement and Research (2022-1-2251), Natural Science Foundation of China (81972148) and Beijing Natural Science Foundation of China (7222101).

**Competing interests** None declared.

**Patient and public involvement** Patients and/or the public were not involved in the design, or conduct, or reporting, or dissemination plans of this research.

**Patient consent for publication** Not applicable.

**Ethics approval** Not applicable.

**Provenance and peer review** Not commissioned; externally peer reviewed.

**Data availability statement** All data relevant to the study are included in the article or uploaded as online supplemental information. This study summarised and analysed the data from the original experiment.

**ORCID iDs**
Conglin Han http://orcid.org/0000-0003-0621-4029
Weijun Gong http://orcid.org/0000-0002-9134-8218

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
