## [Reviewer comments · BMJ Open]

ARTICLE DETAILS

TITLE (PROVISIONAL)	Effects of different aerobic exercises on the global cognitive function of the elderly with mild cognitive impairment: a meta-analysis
AUTHORS	Han, Conglin; Sun, Weishuang; Zhang, Dan; Xi, Xiaoshuang; Zhang, Rong; Gong, Weijun

VERSION 1 – REVIEW

REVIEWER	Wang, Xin Yangzhou University College of Physical Education
REVIEW RETURNED	28-Oct-2022

GENERAL COMMENTS	1. Is the large heterogeneity of some results significant or is it necessary to combine them for analysis?2. Whether specific discussions on future research design suggestions can be added through heterogeneity analysis3. Why not do a net analysis?
--

REVIEWER	Caffo, Alessandro Oronzo Universita degli Studi di Bari Aldo Moro
REVIEW RETURNED	13-Dec-2022

GENERAL COMMENTS	The manuscript titled “Effects of different aerobic exercises on the global cognitive function of the elderly with mild cognitive impairment: a meta-analysis” proposed a systematic review with a formal meta-analysis to evaluate the effect of different types of aerobic exercise on the cognitive functioning in elderly people with Mild Cognitive Impairment (MCI). The global cognitive functioning was analysed through different studies based on the outcome of the simple MMSE or MoCA tests. The aerobic exercise incorporated conventional aerobic exercise, mind-body exercise, and multi-component aerobic exercise. Results were obtained by the search of the relevant literature and 18 articles meeting the inclusion criteria were included. I carefully read the manuscript and I think it could be of interest for the readers of BMJ Open. I also think that some minor points need to be addressed before publication. Below there are my comments and suggestions. This meta-analysis follows the PRISMA guidelines, as stated in the method section. In my opinion it has been conducted accurately, and the authors used proper meta-analytical modeling to address the aim of the study.
--

	In the “Literature Quality Evaluation” section, please specify any parameter of risk of bias assessment that may affect the cumulative estimate in the text. Moreover, the exposition of the different biases is not accurate: please, could you add the categorization for high-risk (line 20, page 4)? The authors are also advised to assess the risk of bias in the included studies using a standardised approach with specific criteria, reporting the results of each assessment. The table is not sufficient because it does not provide complete information on studies with precise methodological limitations. Please, could you include a caption for figure 2, page 16? Finally, it would be recommend changing the name of the section in “Risk of bias assessment”. I also recommend including a funnel plot for the two main meta-analyses, because the number of studies is enough in the first case (i.e., MMSE) and barely sufficient in the second case (i.e., MoCA; it could be reported as a tentative analysis) to test for the eventual impact of publication bias. Indeed, tests for funnel plot asymmetry should be used only when there are at least 10 studies included in the meta-analysis, because when there are fewer studies the power of the tests is too low to distinguish chance from real asymmetry. (Page, Higgins, Stern, 2022). Please, indicate the EMBASE search strategy in ascending order (1 to 22). Discussion and conclusion involve the obtained results considering the relevant literature. I suggest, for the future research, to encompass in the inclusion criteria also studies that employ other tests than MMSE and MoCA to assess cognitive functioning, such as the CDR (Clinical Dementia Rating Scale). Moreover, it would be interesting to include studies that assess specific cognitive domains, as well as memory or attention. This would probably allow to include a higher number of studies and to better disentangle heterogeneity sources among them.
--	--

REVIEWER	Okekunle, Akinkunmi Paul University of Ibadan
REVIEW RETURNED	12-Jan-2023

GENERAL COMMENTS	 • Thank you for the invitation to review this manuscript. Generally, I have reviewed the methods and results of the manuscript. The question of the novelty of this work is a primary concern. How is it different from the several published works in literature (https://pubmed.ncbi.nlm.nih.gov/?term=exercise+and+cognition&filter=pubt.meta-analysis&filter=pubt.systematicreview)? • Besides, the current methods and results of the manuscript are short of critical details necessary to support the art and understanding of the science of the work. I have detailed some of the comments below; Abstract • Lines 46-47: should be deleted. It is a tautology of lines 42-43 Inclusion criteria; • The inclusion criteria did not account for the likely presence of co-morbidity. Older adults are vulnerable high-risk groups to NCDs that are usually treated with pharmacological interventions. So it is uncertain if it is exercise driving the observed association. Did the RCTs document these details that could confound the results? If yes, they should be clearly written and described in the manuscript. If not, this should be a limitation in the manuscript. • Also, were the method of exposure assessment and outcome the same in all the RCTs included in the meta-analysis? That is
--

	unlikely anyway, and subgroup analyses might be necessary to test the viability of the methods in evidence synthesis.  • Was randomization methods and assignment to intervention uniform across studies? • There is no information on the period of the RCTs for inclusion. I am not sure if the authors considered trial durations • Critical details of the methods in the RCTs are necessary to assert the validity of the intervention before doing a meta-analysis. This and other essential information have not been adequately described in the manuscript. Search strategy  • Details of the keywords, MESH terms, hits etc., in each database should be documented in tabular form as supplementary information. I have seen the report for EMBASE, but what of other databases? Data extraction and analysis  • Details of the statistical analysis are missing. The authors mentioned the aggregation of mean differences across studies. How was this done? The principle, basis, assumptions, level of significance, and other vital statistical details should be documented in the manuscript. Literature quality review  • First, what was the basis for the evaluation? What qualifies a study for low, moderate or high risk of bias? • Were there no differences in the assessment if “Two reviewers assessed each trial independently”? If there were. How was this resolved? • I should think there should be a subgroup analysis based on the quality review (low, moderate and high risk) to determine the significance of study quality on the strength of evidence would be necessary. • What of sensitivity analyses? Research characteristics  • Detailed information on the content of the interventions reported in this review might be necessary as supplementary data. E.g., what does the “mind-body exercise intervention” or “mind-body exercise intervention” entail? Supplementary Table 1  • The vital detail missing here is whether there is information on cognitive function at baseline. • Statistical-wise, combining cognitive scores from MMSE and MoCA may not be appropriate. These are two different assessments. It is possible these two scale measures the same thing based on scientific understanding, but statistical-wise, it is unlikely. I would suggest the author consider stratifying the findings based on MMSE or MoCA. Discussion  • The importance or significance, or novelty of this study was not documented • A considerable omission here is the juxtaposition of the results with previous findings. • Also, the statement “We did not find studies on conventional aerobic exercise” might not be entirely accurate. Please review the literature again to assert the opinion. Other comments  • Many results are presented with no detailed information on the methods. • Generally, the current presentation of the result is very tardy. I would suggest the authors consider revising the manuscript considering the thematic representation of results by methods.
--	--

VERSION 1 – AUTHOR RESPONSE

Responses to the comments of Reviewer 1

1. Is the large heterogeneity of some results significant or is it necessary to combine them for analysis?

Response: Thank you very much for your reminder on this issue. I think it makes sense. The results of the MMSE assessment, for example, were significant for all movement types combined, but there was too much heterogeneity. Whereas the results of the separate meta-analysis by movement type showed that the results were significant and less heterogeneous when analyzed separately.

2. Whether specific discussions on future research design suggestions can be added through heterogeneity analysis.

Response: Thank you for your comments, a paragraph has been added to shed light on future research.

3. The reviewer asked why not do a network analysis.

Response: Many thanks! This article is only to explore the effects of different forms of exercise on cognitive function, not to compare which is better.

Responses to the comments of Reviewer 2

1. In the “Literature Quality Evaluation” section, please specify any parameter of risk of bias assessment that may affect the cumulative estimate in the text.

Response: The relevant parameters are referenced in the Cochrane Handbook (Table 8.5.d), and the relevant URLs have been uploaded.

2. Please, could you add the categorization for high-risk (line 20, page 4)?

Response: Thank you very much for your advice. The description of the risk of drift has been refined.

3. The authors are also advised to assess the risk of bias in the included studies using a standardized approach with specific criteria, reporting the results of each assessment.

Response: The relevant standard is referenced in the Cochrane Handbook for Systematic Reviews of Interventions Table 8.7.a. The results are written in section 3.3 Literature quality evaluation.

4. Please, could you include a caption for figure 2, page 16? Finally, it would be recommend changing the name of the section in “Risk of bias assessment”.

Response: Thank you very much for your reminder. Figure 2 is named Figure 2: Risk of bias assessment of included trials. The name of the section in “Risk of bias assessment” has been changed to “Risk of Bias and quality assessment”.

5. I also recommend including a funnel plot for the two main meta-analyses.

Response: The reasons for not adding a funnel plot are that the focus of this paper is on meta-analysis results of different modes of motion, with a small number of studies for each mode of movement (≤ 5), and because some of the results are too heterogeneous, which may cause inaccurate funnel plots.

6. Please, indicate the EMBASE search strategy in ascending order (1 to 22).

Response: Many thanks for your suggestion. The EMBASE search strategy has been changed to ascending order.

7. I suggest, for the future research, to encompass in the inclusion criteria also studies that employ other tests than MMSE and MoCA to assess cognitive functioning, such as the CDR (Clinical Dementia Rating Scale). Moreover, it would be interesting to include studies that assess specific cognitive domains, as well as memory or attention. This would probably allow to include a higher number of studies and to better disentangle heterogeneity sources among them.

Response: Thank you so much for your advice. We have modified it accordingly.

Responses to the comments of Reviewer 3

Abstract:

1. Lines 46-47: should be deleted. It is a tautology of lines 42-43.

Response: The duplicates have been removed.

Inclusion criteria:

1. The inclusion criteria did not account for the likely presence of co-morbidity. Older adults are vulnerable high-risk groups to NCDs that are usually treated with pharmacological interventions. So it is uncertain if it is exercise driving the observed association. Did the RCTs document these details that could confound the results? If yes, they should be clearly written and described in the manuscript. If not, this should be a limitation in the manuscript.

Response: Thank you very much for asking this interesting question. We took comorbidities into account when screening the study, and the criteria for inclusion essentially excluded subjects whose experimental results were affected by their condition (e.g., subjects taking medications that might affect cognitive function, and people with medical conditions that might affect doing aerobic exercise). Nonetheless, we consider your suggestions to be very important and have enriched this manuscript.

2. Were the method of exposure assessment and outcome the same in all the RCTs included in the meta-analysis? That is unlikely anyway, and subgroup analyses might be necessary to test the viability of the methods in evidence synthesis.

Response: This study analyzed the impact of different types of aerobic exercise on global cognitive impairment. Due to the limited number of included literature, no subgroup analysis was performed on specific cognitive domains. We will pay attention to this in future research.

3. Were randomization methods and assignment to intervention uniform across studies?

Response: This paper included 18 RCT studies, the randomization method of 13 studies was simple randomization, and 5 studies were stratified randomization. In terms of intervention allocation concealment, 9 studies used open random tables, 3 studies used sealed envelopes, and the remaining 6 studies were unclear.

4. There is no information on the period of the RCTs for inclusion.

Response: Many thanks! We have added this part to our manuscript.

5. Critical details of the methods in the RCTs are necessary to assert the validity of the intervention before doing a meta-analysis.

Response: The inclusion criteria have been restated.

Search strategy:

1. Details of the keywords, MESH terms, hits etc., in each database should be documented in tabular form as supplementary information. I have seen the report for EMBASE, but what of other databases?

Response: Search strategies for PubMed and the Cochrane Library have been uploaded as supplementary material.

Data extraction and analysis:

1. Details of the statistical analysis are missing. The authors mentioned the aggregation of mean differences across studies. How was this done? The principle, basis, assumptions, level of significance, and other vital statistical details should be documented in the manuscript.

Response: We apologize for the lack of detail in this part of the previous manuscript. We have revised the Statistical Methods section to add details of the statistical analysis.

Literature quality review:

1. What was the basis for the evaluation? What qualifies a study for low, moderate or high risk of bias?

Response: The relevant standard is referenced in the Cochrane Handbook for Systematic Reviews of Interventions Table 8.7.a, and the relevant URLs have been uploaded.

2. Were there no differences in the assessment if "Two reviewers assessed each trial independently"? If there were. How was this resolved?

Response: Many thanks! We have added this part to our manuscript.

3. I should think there should be a subgroup analysis based on the quality review (low, moderate and high risk) to determine the significance of study quality on the strength of evidence would be necessary.

Response: Due to the small number of studies (≤ 5) for each exercise type, we did not perform subgroup analyses by low, intermediate, and high risk. Instead, subgroup analyses were performed within exercise types with higher heterogeneity.

4. What of sensitivity analyses?

Response: Thank you very much for your reminder on this issue. We performed sensitivity analyses separately in the two groups with high heterogeneity of results.

Research characteristics:

1. Detailed information on the content of the interventions reported in this review might be necessary as supplementary data. E.g., what does the “mind-body exercise intervention” or “mind-body exercise intervention” entails?

Response: We have added details of the interventions to the inclusion criteria.

Supplementary Table 1:

1. The vital detail missing here is whether there is information on cognitive function at baseline.

Response: We have supplemented the baseline information.

2. Statistical-wise, combining cognitive scores from MMSE and MoCA may not be appropriate. These are two different assessments. It is possible these two scale measures the same thing based on scientific understanding, but statistical-wise, it is unlikely. I would suggest the author consider stratifying the findings based on MMSE or MoCA.

Response: Thank you so much for your advice. In this study, we did not combine the cognitive scores of MMSE and MoCA for analysis but stratified the findings based on these two indicators.

Discussion:

1. The importance or significance, or novelty of this study was not documented. A considerable omission here is the juxtaposition of the results with previous findings.

Response: Aerobic exercise is the most common form of exercise in life, and we think it is important to classify and discuss it. Where previous studies have only analyzed aerobic exercise in general or other forms of exercise, this study breaks down the various forms of aerobic exercise. Although the main results do not differ much from those of previous studies, some novelties were found. For example, despite the positive effects of multi-component aerobic exercise, the results are not consistently reliable and may require further investigation.

2. The statement “We did not find studies on conventional aerobic exercise” might not be entirely accurate. Please review the literature again to assert the opinion.

Response: The original sentence has been corrected.

Other comments:

1. Many results are presented with no detailed information on the methods.

Response: Detailed information on the methodology has been added to the manuscript.

2. Generally, the current presentation of the result is very tardy. I would suggest the authors consider revising the manuscript considering the thematic representation of results by methods.

Response: Thank you so much for your advice. As a preliminary exploration of the effects of different types of aerobic exercise on cognitive function, this study has obtained some guiding results. In further research, we consider using different methods to explore the relevant topics.

3. A language editorial review might be necessary as well.

Response: We thank the reviewer for the comment and we have double-checked the language throughout the manuscript.

VERSION 2 – REVIEW

REVIEWER	Wang, Xin Yangzhou University College of Physical Education
REVIEW RETURNED	24-Feb-2023

GENERAL COMMENTS	This is a good article about MCI.
-----------------------------------

REVIEWER	Alessandro Oronzo Universita degli Studi di Bari Aldo Moro
REVIEW RETURNED	09-Mar-2023

GENERAL COMMENTS	I carefully read the revised version of the manuscript, and I think that Authors have addressed all the issues raised by the reviewers. In my opinion, the manuscript could be accepted for publication.
--

REVIEWER	Okekunle, Akinkunmi Paul University of Ibadan
REVIEW RETURNED	07-Mar-2023

GENERAL COMMENTS	Kudos for the efforts invested in the revision of the manuscript. I have read the revised manuscript again and think the authors have made some significant revisions to the manuscript. However, the authors did not effectively detail the changes, with no clear-cut emphasis on how these changes affected the revised manuscript. Also, the authors did not itemize the revision details in reply to earlier comments. The specific lines and pages where the changes were effected in the revised manuscript were not included in the reply to the reviewer. This has made efforts to review the authors' noble work quite tedious. A. In specific terms, to itemize some unaddressed comments from the authors' replies; >Data extraction and analysis: 1. Details of the statistical analysis are missing. The authors mentioned the aggregation of mean differences across studies. How was this done? The principle, basis, assumptions, level of significance, and other vital statistical details should be documented in the manuscript. Response: We apologize for the lack of detail in this part of the previous manuscript. We have revised the Statistical Methods section to add details of the statistical analysis." This is still yet to be included in the statistical analysis. The underlying assumptions guiding the meta-analysis were not reported. A random effect model was reported in the abstract, but the modus operandi and it assumptions were omitted in the methods. "Discussion 1. The importance or significance, or novelty of this study was not documented. A considerable omission here is the juxtaposition of the results with previous findings. Response: Aerobic exercise is the most common form of exercise in life, and we think it is important to classify and discuss it. Where previous studies have only analyzed aerobic exercise in general or other forms of exercise, this study breaks down the various forms of aerobic exercise. Although the main results do not differ much from those of previous studies, some novelties were found. For example, despite the positive effects of multi-component aerobic exercise, the results are not consistently reliable and may require further investigation. " Again, this was informative but not included in the revised manuscript. I advise the authors to simplify this response
---

	statement and have it in the first paragraph of the Discussion to itemize the study's novelty. "Other comments: 1. Many results are presented with no detailed information on the methods. Response: Detailed information on the methodology has been added to the manuscript." Also, here the authors asserted some additional information had been documented in the methods without detailing the findings in the result section. "2. Generally, the current presentation of the result is very tardy. I would suggest the authors consider revising the manuscript considering the thematic representation of results by methods. Response: Thank you so much for your advice. As a preliminary exploration of the effects of different types of aerobic exercise on cognitive function, this study has obtained some guiding results. In further research, we consider using different methods to explore the relevant topics" Also, my earlier comment in this regard was to itemize the need to present the results in well-articulated and synchronous detail in describing the study results. The authors should consider revising the results. Other additional comments; B. A subgroup analysis by risk of bias and quality assessment might be necessary to test the strength of evidence by quality of studies. These results should be detailed in the supplement and briefly described in the methods section C. The sensitivity analysis is incomplete as it should have been conducted by figures 3, 4, 5&6 D. Also, the sensitivity analyses should be constricted to one table. I will be happy to review the revision again and kindly urge the authors to detail the changes made (including the lines and page numbers) in reply to the reviewer's comments. Thank you.
--	---

VERSION 2 – AUTHOR RESPONSE

Responses to the comments of Reviewer 3

"Data extraction and analysis:

1. Details of the statistical analysis are missing. The authors mentioned the aggregation of mean differences across studies. How was this done? The principle, basis, assumptions, level of significance, and other vital statistical details should be documented in the manuscript.

Response: We apologize for the lack of detail in this part of the previous manuscript. We have revised the Statistical Methods section to add details of the statistical analysis."

This is still yet to be included in the statistical analysis. The underlying assumptions guiding the meta-analysis were not reported. A random effect model was reported in the abstract, but the modus operandi and its assumptions were omitted in the methods.

Response : Thank you very much for your reminder on this issue. We have added relevant content to the manuscript.(Page 3, lines 14-16)

"Discussion

1. The importance or significance, or novelty of this study was not documented. A considerable omission here is the juxtaposition of the results with previous findings.

Response: Aerobic exercise is the most common form of exercise in life, and we think it is important to classify and discuss it. Where previous studies have only analyzed aerobic exercise in general or other forms of exercise, this study breaks down the various forms of aerobic exercise. Although the main results do not differ much from those of previous studies, some novelties were found. For example, despite the positive effects of multi-component aerobic exercise, the results are not consistently reliable and may require further investigation. "

Again, this was informative but not included in the revised manuscript. I advise the authors to simplify this response statement and have it in the first paragraph of the Discussion to itemize the study's novelty.

Response : Thank you so much for your advice. We have modified the first paragraph of the discussion.(Page 6, lines 58-62)

"Other comments:

1. Many results are presented with no detailed information on the methods.

Response: Detailed information on the methodology has been added to the manuscript."

Also, here the authors asserted some additional information had been documented in the methods without detailing the findings in the result section.

Response : Detailed information about the partial results has been added to the method (Page 3, lines 8-11, 31-34).

2. Generally, the current presentation of the result is very tardy. I would suggest the authors consider revising the manuscript considering the thematic representation of results by methods.

Response: Thank you so much for your advice. As a preliminary exploration of the effects of different types of aerobic exercise on cognitive function, this study has obtained some guiding results. In further research, we consider using different methods to explore the relevant topics"

Also, my earlier comment in this regard was to itemize the need to present the results in well-articulated and synchronous detail in describing the study results. The authors should consider revising the results.

Response : Many thanks for your suggestion. The presentation of the results has been modified as you suggested (Page 5-6 , the modified parts are marked in green).

Other additional comments;

B. A subgroup analysis by risk of bias and quality assessment might be necessary to test the strength of evidence by quality of studies. These results should be detailed in the supplement and briefly described in the methods section

Response : We performed relevant subgroup analyses and assessed the strength of the evidence(Page 3, lines 31-34. Page 4, lines 6-8. Page 5, lines 47-52. Supplementary Document 3&5).

C. The sensitivity analysis is incomplete as it should have been conducted by figures 3, 4, 5&6

D. Also, the sensitivity analyses should be constricted to one table.

Response : Thank you very much for your advice. The content about sensitivity analysis has been revised according to your suggestion(Page 6, lines 50-54, Supplementary Document 6).

VERSION 3 – REVIEW

REVIEWER	Okekunle, Akinkunmi Paul University of Ibadan
REVIEW RETURNED	22-May-2023
GENERAL COMMENTS	The authors have addressed all comments, and I have no further comments. Thank you.